# Intelligent Poly(*N*-Isopropylmethacrylamide) Hydrogels: Synthesis, Structure Characterization, Stimuli-Responsive Swelling Properties, and Their Radiation Decomposition

**DOI:** 10.3390/polym12051112

**Published:** 2020-05-13

**Authors:** Snežana Ilić-Stojanović, Maja Urošević, Ljubiša Nikolić, Djordje Petrović, Jelena Stanojević, Stevo Najman, Vesna Nikolić

**Affiliations:** 1Faculty of Technology, University of Niš, Bulevar Oslobodjenja 124, 16000 Leskovac, Serbia; maja@tf.ni.ac.rs (M.U.); nljubisa@tf.ni.ac.rs (L.N.); jstanojevic@tf.ni.ac.rs (J.S.); nikolicvesna@tf.ni.ac.rs (V.N.); 2Vinča Institute of Nuclear Sciences, Department of Radioisotopes, University of Belgrade, Mike Petrovića Alasa 12-14, 11351 Vinča, Belgrade, Serbia; djpetrovic@vin.bg.ac.rs; 3Faculty of Medicine, University of Niš, Boulevard dr Zorana Djindjica 81, 18108 Niš, Serbia; stevo.najman@medfak.ni.ac.rs

**Keywords:** *N*-isopropylmethacrylamide, hydrogel, swelling kinetics, gamma irradiation, decrosslinking

## Abstract

Poly(*N*-isopropylmethacrylamide) (p(*N*iPMAm)) is one of the lesser known homopolymers that has significant potential for designing new “intelligent” materials. The aims of this work were the synthesis a series of cross-linked p(*N*iPMAm) hydrogels by the free radical polymerization method and the application of gamma-ray radiation for additional cross-linking. The synthesized p(*N*iPMAm) hydrogels were structurally characterized by Fourier transform infrared spectroscopy (FTIR) and scanning electron microscopy (SEM). The amount of unreacted monomers was analyzed using high pressure liquid chromatography (HPLC) to evaluate conversion of monomers into polymers. The swelling behavior was monitored in dependence of pH and temperature changes. The previous aim of gamma-ray radiation was the further the cross-linkage of the obtained hydrogel sample in the gelatinous, paste-like state, but the gamma-ray radiation caused decomposition. After absorbing irradiation doses, they transformed into the liquid phase. The results obtained by the gel permeation chromatography (GPC) method indicated that only oligomers and monomers were present in the irradiated liquid material, without molecules with a higher average molar mass, i.e., that the decomposition of the hydrogels occurred. Additionally, the irradiated liquid material was analyzed using the static headspace gas chromatography mass spectrometry (HSS-GC/MS) and gas chromatography/flame ionization detection (HSS-GC/FID) methods. The presence of unchanged initiator molecule and a dominant amount of four new molecules that were different from homopolymers and the reactant (monomer and cross-linker) were determined.

## 1. Introduction

Responsive, or stimuli-sensitive, hydrogels are cross-linked three-dimensional polymeric matrices based on natural or synthetic materials that have the ability to swell while retaining a large amount of surrounding fluid, but they do not dissolve [1]. They are known as “intelligent” or “smart” hydrogels, because when environmental factors change (e.g., pH-value, temperature, light, glucose concentration, pressure, and magnetic or electric fields), they can respond with the changes of some physicochemical property (e.g., swelling capacity) [2]. Their syntheses can be designed to vary swelling ability, porosity, physical structure, etc., in order to provide an adequate response to the external stimuli changes [3]. All changes that occur in the hydrogels are reversible, and they can return to their original state upon the removal of the stimulus. Thermosensitive hydrogels with an upper critical solution temperature (UCST) swell above and contract below, while polymers with a lower critical solution temperature (LCST) swell below and contract over critical temperature [2,4].

Poly(*N*-isopropylmethacrylamide), p(*N*iPMAm), has significant potential for designing new materials. Research on this homopolymer and its copolymers has been largely overshadowed by the many scientific studies of the more popular poly(*N*-isopropylacrylamide), p(*N*iPAm), representative [5]. The p(*N*iPMAm) belongs to the group of thermosensitive hydrogels [6], and possesses an additional methyl group on each monomer unit related to p(*N*iPAm) [7]. The phase transition of aqueous solution of p(*N*iPMAm) has been analyzed by several authors in the last two decades [8,9]. It is assumed that methyl groups are responsible for higher LCST in relation to p(*N*iPAm). Because of the steric hindrance effect due to the presence of an additional methyl group, hydrophobic side groups cannot interact with each other, causing the phase transition to occur at a higher temperature in the range of 38–42 °C [9,10]. The p(*N*iPMAm) forms less stable aggregates during the phase transition that can be almost completely dehydrated above the LCST, compared to the p(*N*iPAm) behavior [11]. An analysis of the association and dissociation of p(*N*iPMAm) chains in water showed that p(*N*iPMAm) chains form larger aggregates at temperatures above LCST and have less enthalpy and entropy changes, as well as smaller conformational changes, than p(*N*iPAm) [12]. Through a quantitative comparison of the reversibility of temperature-induced hydration of p(*N*iPMAm) and p(*N*iPAm) in aqueous solutions by 2D correlation analysis, it was found that the hydration of the hydrophilic amide group has a dominant influence on the temperature-induced polymer separation [13]. Though p(*N*iPMAm) has a high LCST value, the phase transition is very sharp, making it an adequate monomer for copolymerization compared to p(*N*iPAm) [14,15]. Based on the swelling data of the thermo- and pH-responsive copolymers based on *N*-isopropylmethacrylamide (*N*iPMAm) and *N*-isopropylacrylamide (*N*iPAm), acrylic acid, itaconic acid, and sodium methacrylate, a model for mechanical behavior and water absorption was developed [16]. A comparative analysis of p(*N*-*n*-propylacrylamide), p(*N*iPMAm) and p(*N*iPAm) microgels synthesized by conventional surfactant-free precipitation polymerization showed that p(*N*iPMAm) particles have a more homogeneous matrix compared to the other two systems [17]. The colloidal and physicochemical characteristics of cross-linked p(*N*iPMAm) latex microgel were found to exhibit a thermo-sensitivity because the particle diameter decreases after increasing the temperature, with an LCST value of about 43 °C [18]. The deuteration-induced volume phase transition temperature shift of p(*N*iPMAm) microgels was studied [19]. A swelling test of p(*N*iPMAm) and p(*N*iPAm) in water/ethanol mixtures at room temperature manifested a strong cononsolvency effect occurring on the uncharged and negatively charged hydrogels. The higher swelling degree was observed for pure solvents, and minimal swelling was observed for solvent mixtures [20]. The additional methyl group in the p(*N*iPMAm) chains slightly affected the kinetics of the helix transition to the globule of the linear chains. A comparison of the phase transition temperature in aqueous solution p(*N*iPAm) and p(*N*iPMAm) showed that the presence or absence of an α-methyl group had a strong influence on the physical structure of the solution. Though the hydrophobic interactions of p(*N*iPMAm) and p(*N*iPAm) are very similar, p(*N*iPMAm), with the additional methyl group, exhibits significantly weaker intermolecular interactions between the amide group. This is the reason that the p(*N*iPMAm) phase transition temperature has been found to increase by about 8 °C compared to that of p(*N*iPAm) [21]. The influence of the α-methylation on the main chain of both the proton donor and proton acceptor polymers, as well as on the interpolymeric complexes between p(*N*iPMAm) and p(*N*iPAm) with poly(acrylic acids) and poly(methacrylic acids), was studied. The effect of α-methyl substitution on the interpolymer complex showed that α methylation provides higher glass transition temperatures (168–228 °C) and delays of thermal degradation [22]. The structure of doubly-temperature sensitive core-shell microgels based on p(*N*iPMAm) and p(*N*iPAm) was examined [23,24]. The swelling and mechanical behavior of negatively charged p(*N*iPMAm) hydrogels, depending on temperature in water and in aqueous NaCl solutions at room temperature, were investigated [25]. The volume phase transition temperatures and swelling behaviors of *N*-isopropylmethacrylamide-*co-N*-isopropylacrylamide gels with various mole ratios prepared by precipitation polymerization were studied [26].

The synthesis of poly(acrylamides) can be performed using many techniques. Common methods for the synthesis of hydrogels include radical polymerization with thermal [8,9], chemical [12,27], UV, or radiation initiation by the action of a microwave field or the use of a suitable initiator [28]. P(*N*iPMAm) is usually obtained by standard radical polymerization, which cannot control the tacticities of the resulting polymer [29]. Poly(*N*-isopropylmethacrylamide-*co*-itaconic acid) gels prepared by free-radical cross-linking copolymerization in various conditions and swelling characteristics were described [3]. The surfactant-free, radical precipitation copolymerization of *N*iPMAm and the cationic *N*-(3-aminopropyl)methacrylamide hydrochloride was carried out to prepare functionalized microgels [30]. The p(*N*iPMAm) microgels were prepared by precipitation polymerization using the anionic and the cationic surfactant to precisely control the size of microgels [31]. There are a lot of types of hydrogels that are mostly prepared by inverse suspension, emulsion polymerization, and solution polymerization [29]. However, in most cases, these polymerization techniques require special reaction conditions, and they demonstrate difficult yielding lower polydispersities and high molecular weights. Because of that, interest in the design, synthesis, and application of “intelligent” polymers in such forms as linear, grafted, or cross-linked chains is increasing.

Radiation polymerization was recognized a long time ago as a powerful technique for the formation of hydrogels [32]. The main advantage of this technique is that it is a cold process and does not require the use of any initiators or cross-linkers, which are known for their high cost and toxic nature [33]. It is used for the simultaneous synthesis, modification, and sterilization of polymers [34]. Gamma-radiation can induce various effects, e.g., cross-linking or scission in polymers and networks, polymerization and grafting of natural/synthetic polymers, dye discoloration, pesticide degradation, and the chemical activation of organic materials by oxidation [35].

Radiation-induced cross-linking changes linear polymers into 3D molecules due to intermolecular cross-linking, which increases the average size and molecular weight of the starting polymer [36]. Chain reactions during the polymerization process can usually be achieved with medium gamma-ray absorbed doses. The processes involving single steps or short kinetic chain length reactions typically require higher doses, as in the case of the radiation cross-linking of rubbers and thermoplastics [34]. The radiation-modification of chitosan with *N,N*-dimethylacrylamide results in different architectures (grafting hydrogels, interpenetrating, and semi-interpenetrating networks) with a porous and interconnected structures were achieved by gamma irradiation dose of 6 kGy by a ^60^Co source [37]. *N*-isopropylacrylamide was successfully grafted onto chitosan via γ-radiation at a dose of 3–10 kGy [38]. The structural characteristics and bonding environment of Ag nanoparticles synthesized by gamma irradiation within thermo-responsive p(*N*iPAm) polymer composites were investigated [39]. A new superabsorbent hydrogel for diaper applications based on Tara gum and acrylic acid was synthesized by 14.5 kGy gamma irradiation in air at ambient temperature [40]. At lower doses, the most significant reduction in the viscosity of samples was at the early stages of irradiation. The rate of viscosity reduction is lower for polymers irradiated in a solid than irradiated in a solution, which proves that the presence of water accelerates radiation-induced reactions via an indirect effect, through intermediate products of water radiolysis, with the scission of the main chain. The reduction of solution viscosity is directly related to molecular weight.

However, ionizing radiation causes main chain scission, fragmentation and ring-opening reactions. Experimental analyses have shown that kappa-carrageenan (KC) degrades after irradiation, γ-irradiation destroys the main chains of KC and the hydrogen bonds between KC and water molecules in KC gels. The cross-linking reaction of poly(*N*-vinylpyrrolidone) (PVP) has been shown to be quicker than the degradation of KC at a low dose (less than 30 kGy), and KC degradation is inhibited in a mixture with PVP [41]. Hydrogel synthesis upon the irradiation of polymer in solution may occur basically through intramolecular reactions and the intermolecular cross-linking of polymer radicals [42]. Since there is always a competition between these two processes and degradation, the ultimate goal is to choose the right conditions that promote intramolecular and intermolecular cross-linking and reduce the probability of undesirable processes.

In a previous publication, a literature review of hydrogels based on p(*N*iPMAm) and p(*N*iPAm) was described [43]. The similarity of the thermosensitive monomers of *N*-isopropylmethacrylamide and *N*-isopropylacrylamide allows their copolymers to achieve a sharp phase transition under physiological conditions. Recently, this research team published an article about copolymeric poly(*N*-isopropylmethacrylamide-*co*-*N*-isopropylacrylamide) hydrogels synthesized by radical polymerization [44]. These negatively thermosensitive cross-linked p(*N*iPMAm/*N*iPAm) hydrogels were structurally characterized, and their swelling behavior was investigated in relation to the temperature and pH value of their solution. The aims of this work were the synthesis a series of cross-linked homo-polymeric hydrogels p(*N*iPMAm) by a free radical polymerization method, as well as their characterization and the application of gamma-ray radiation in order to further the initiation of additional cross-linking. The structural characterization of hydrogels performed by the Fourier transform infrared spectroscopy (FTIR) and scanning electron microscopy (SEM) methods and their swelling behavior from the aspect of pH and thermal sensitivity were analyzed for the further discovery of their possible applications. The gamma-ray radiation caused the radiolysis of a gelatinous, paste-like hydrogel sample instead of the formation of a firmer three-dimensional structure. Degradation was confirmed using gel permeation chromatography (GPC) and the static headspace gas chromatography mass spectrometry (HSS-GC/MS) and gas chromatography/flame ionization detection (HSS-GC/FID) methods.

## 2. Materials and Methods

### 2.1. Materials

Our materials included *N*-isopropylmethacrylamide, *N*iPMAm, 97% (Acros Organics, Morris Plains, NJ, USA) used without further purification, 97% ethylene glycol dimethacrylate (EGDM) (Fluka Chemical Corp, Buchs, Switzerland), 98% 2,2′-azobis(2-methylpropionitrile) (AZDN) (Acros Organics, Morris Plains, NJ, USA), 99.5% ethanol (Merck KGaA, Darmstadt, Germany), and methanol for high pressure liquid chromatography (HPLC), gradient grade, ≥99.9% (Merck KGaA, Darmstadt, Germany).

### 2.2. Method of p(NiPMAm) Hydrogel Synthesis

The samples of homopolymer p(*N*iPMAm) were synthesized using the free radical polymerization method of the *N*iPMAm monomer with 1.5, 2.0, 2.5, and 3.0 mol% of EGDM as cross-linker. The polymerization reaction was initiated by adding of the 2.8 mol% of AZDN as an initiator. Ethanol was applied as a solvent. The reaction mixtures were homogenized and injected into glass ampoules, which were then sealed. The polymerization reaction for all samples was thermally initiated according to the following regime: 30 min at 70 °C, 120 min at 80 °C, and 30 min at 85 °C. After cooling, the synthesized p(*N*iPMAm) hydrogels were separated from the glass ampoules in the form of long cylinders and cut into smaller cylinders (*d*⋅*l* = 5⋅2, where *d* is the diameter in mm and *l* is the thickness after drying in mm). The hydrogels were then treated with methanol for 36 h (0.5 g of hydrogel was overflowed with 30 cm^3^ of methanol) as to remove all unreacted compounds. After methanol treatment, the hydrogels were immersed in the methanol/distilled water solutions 75%/25%, 50%/50%, 25%/75%, and 0%/100%, *v*/*v*, after 24 h to gradually rinse the methanol from synthesized hydrogels. The hydrogels were dried for about 3 h at 40 °C to the constant mass. After drying, the obtained p(*N*iPMAm) xerogels were stored in a desiccator and subjected to further analysis.

### 2.3. Structural Confirmation of p(NiPMAm) Hydrogels

#### 2.3.1. Fourier Transform Infrared Spectroscopy

The FTIR spectrum of the EGDM cross-linker was recorded using a capillary film method between two polished CaF_2_ plates. The FTIR spectra of *N*iPMAm monomers and p(*N*iPMAm) synthesized xerogels were prepared using the KBr pellet method. For the preparation of thin transparent pellets, 150 mg potassium bromide with spectroscopic purity and 0.9 mg of samples were measured and then ground to powder state in an amalgamator (WIG-L-BVG, 31210-3A). Transparent pellets were subjected to vacuuming and pressing under a pressure of about 200 MPa. The recordings were performed in the wavenumber range of 4000–400 cm^−1^ on a Bomem Hartmann and Braun MB-series FTIR spectrophotometer (Hartmann and Braun, Baptiste, Quebec, QC, Canada). The FTIR spectra were processed using the Win-Bomem Easy software.

#### 2.3.2. Scanning Electron Microscopy Analysis

Scanning electron microscopy (SEM) was used to examine the morphology of the synthesized p(*N*iPMAm) hydrogels. The homopolymer samples p(*N*iPMAm) in the equilibrium swelling state were lyophilized on an Edwards, Mini Fast 680 laboratory freeze-dryer (Edwards Ltd, Eastbourne, UK). The lyophilized hydrogel samples were immersed into nitrogen before cutting to prevent breakage and deformation. After that, the samples were sprayed by an alloy of gold and palladium (85%/15%) under vacuum in a Fine Coat JEOL JFC-1100 Ion Sputter (JEOL Co., Tokyo, Japan). Metalized p(*N*iPMAm) hydrogel samples were scanned with a JEOL Scanning Electron Microscope JSM-5300 (JEOL Co., Tokyo, Japan).

#### 2.3.3. Swelling Study

A gravimetric method was used to measure the equilibrium swelling ratio of the p(*N*iPMAm) hydrogels. The equilibrium swelling weights were measured at 20 °C for the hydrogel samples in a solution of an appropriate pH value (2.0 and 7.4) after wiping excess water from the hydrogel surface with moistened filter paper. The masses of the samples were measured before starting and at defined intervals until equilibrium was reached, i.e., a constant sample mass was achieved.

The temperature sensitivity on the equilibrium swelling ratio of p(*N*iPMAm) hydrogels was measured at the temperature range of 20–60 °C. At each particular temperature, hydrogel samples were incubated in deionized water for 24 h, wiped with moistened filter paper to remove excess water from the hydrogel surface, and weighed.

The swelling ratio, α, and the equilibrium swelling ratio, α_e_, were calculated according to Equations (1) and (2), respectively:(1)$$\alpha =\frac{m-m_{0}}{m_{0}}$$
where *m*_0_ is the xerogel mass and *m* is the mass of the swollen hydrogel at the time *t*.
(2)$$\alpha_{e}=\frac{m_{e}-m_{0}}{m_{0}}$$
where *m*_e_ is the mass of the swollen hydrogel at equilibrium state.

#### 2.3.4. Residual Reactants Analysis

The residual amounts of the unreacted *N*iPMAm monomer in the samples after p(*N*iPMAm) hydrogel synthesis were determined with the HPLC method. All methanol extracts, after synthesized hydrogel treatment during 36 h with stirring, were filtered through a 0.45 μm cellulose membrane filter. The analysis were made on an HPLC Agilent 1100 Series apparatus with a 1200 Series diode-array detector (DAD) (Agilent Technologies, Santa Clara, CA, USA). The detector was tuned to a wavelength of 212 nm. The ZORBAX Eclipse XDB-C18 column (4.6 mm × 250 mm, 5 μm, Agilent Technologies, Santa Clara, CA, USA) was thermostated at 25 °C. The mobile phase, i.e., the eluent was methanol for HPLC at the flow rate 1 cm^3^/min. The injected sample volume was 10 μL for analysis. In order to construct the calibration curve, the *N*iPMAm standard substance for preparation a series of standard solutions with known concentrations was applied. All solutions were filtered through a cellulose membrane filter and analyzed by the HPLC method. The standard substance *N*iPMAm was used for the calibration curve. A series of *N*iPMAm standard solutions with known concentrations were prepared, filtered through a 0.45 μm cellulose membrane filter, and analyzed by the HPLC method. The calibration curve for *N*iPMAm was linear for peak areas in the range of 0.005–0.408 mg/cm^3^, and for this linearity, Equation (3) applies:(3)A=807.753+36878.27⋅c
where *A* is the peak area (mAU⋅s) and *c* is the content of *N*iPMAm (mg/cm^3^), from which the unknown concentration of *N*iPMAm was calculated. From the peaks integration data of the tested methanol extract samples, the obtained peak area values were in the range of the calibration curve.

#### 2.3.5. Gamma-Ray Irradiation of Hydrogel

The p(*N*iPMAm) hydrogel sample with 1.5 mol% of EGDM, which was in gelatinous, paste-like state, was subjected to a γ-ray source, firstly with medium gamma-ray absorbed doses, based on analyzed literature [37,38,39].

The prepared samples were placed in a hot cell at the Laboratory for Radioisotopes, Vinča Institute of Nuclear Sciences, RS, and irradiated during 24 h with dose rate of 250 Gy/h using spent ^192^Ir and ^75^Se that originated from gamma cameras used in radiography. However, after the absorption of a total radiation dose of 6 kGy, the partial dissolution of the gelatinous hydrogels was achieved.

Thereafter, for the activation of reactive species, the hydrogel samples were subjected to higher radiation doses for the purpose of radiation-induced intramolecular and intermolecular cross-linking [39]. In the second phase, the p(*N*iPMAm) sample was irradiated with a source ^60^Co at the Radiation Unit for Industrial Sterilization and Conservation, Vinča Institute of Nuclear Sciences, RS, with dose of 25 kGy.

Surprisingly, after the higher radiation dose absorption during the second irradiation process, the partially dissolved p(*N*iPMAm) hydrogel resulted in a complete transition to the liquid state. The obtained irradiated liquid material from p(*N*iPMAm) hydrogel was subjected to further methods (GPC, HSS-GC/MS, and HSS-GC/FID) for the analysis of the composition of the resulting liquid material.

#### 2.3.6. Gel Permeation Chromatography Analysis of Hydrogel after Radiolysis

Gel permeation chromatography (GPC) was applied to separate the fractions by molecule size in polymers in order to provide the molecular weight distribution and to determine the molar mass of the liquid material obtained after p(*N*iPMAm) hydrogel irradiation. GPC analysis was performed on an Agilent 1200 Series device with a refractive index detector (RID 1200) (Agilent Technologies, Santa Clara, CA, USA). The Zorbax Eclipse PSM 60 and 300 columnns in series (6.2 mm × 250 mm, 5 μm, linear *M*w operating range up to 300,000 g/mol, Agilent Technologies, Santa Clara, CA, USA) was thermostated at 25 °C. The mobile phase was redistilled water with a flow rate of 1 cm^3^/min, and the volume of injected specimens was 10 µl. The ChemStation GPC Data Analysis Software (Agilent GPC-Addon) was used to process the data.

The *N*iPMAm standard solutions, sulfonated poly(styrene), and irradiated liquid polymeric material from p(*N*iPMAm), all at 2 mg/cm^3^, were prepared for GPC analysis by dissolving in redistilled water and filtered on a cellulose membrane filter with a pore diameter of 0.45 μm. The standards of *N*iPMAm and sulfonated poly(styrene) homopolymers with very narrow molecular weight distributions of 12,523, 63,158, and 300,000 g/mol were used for calibration curve construction.

#### 2.3.7. Static Headspace Gas Chromatography Mass Spectrometry and Gas Chromatography/Flame Ionization Detection Analysis of Hydrogel after Radiolysis

An adequate amount of irradiated liquid polymeric material from p(*N*iPMAm) was put in a headspace vial of 20 cm^3^, closed with aluminum crimp cap with a polytetrafluoroethylene/silicone septum (PTFE/Si, 20 mm, Agilent Technologies, Santa Clara, CA, USA) and placed in an HS autosampler (Agilent 7697A Headspace Sampler) of a gas chromatograph (Agilent Technologies 7890B) equipped with an inert, mass selective detector (Agilent MSD 5977A). Chromatography was achieved by using weakly polar, silica capillary HP-5MS (5% diphenyl- and 95% dimethyl-polysiloxane, 30 m × 0.25 mm, 0.25 μm film thickness; Agilent Technologies, Santa Clara, CA, USA) column. Helium was used as the carrier gas at a constant flow rate of 1 cm^3^/min. The parameters of HS autosampler were set as follows: oven temperatures were set at 60, 90, 120, and 150 °C, while the temperatures of the loop and transfer line were 160 and 180 °C, respectively. The vial was equilibrated for 15 min, and the injection duration was 0.50 min. The gas phase was injected in the GC column in the splitless mode. A temperature program was started at 40 °C, where it was held for 4 min, then increased to 260 °C at the rate of 5 °C/min, and finally increased to 290 °C at the rate of 30 °C/min, where it was held for 5 min. Separated components were analyzed with a mass spectrometer. The temperatures of the MSD transfer line, ion source, and quadruple mass analyzer were set at 300, 230, and 150 °C, respectively. The applied ionization voltage was 70 eV, and mass detection was carried out in scan mode in *m*/*z* ranges from 35 to 650.

Quantitative results were obtained on the gas chromatograph used for the GC/MS analysis that was equipped with the flame-ionization detector (FID) of the same company. The GC program was the same as previously described for GC/MS analysis. The flows of the carrier gas (He), make up gas (N_2_), fuel gas (H_2_), and oxidizing gas (Air) were 1, 25, 30, and 400 cm^3^/min, respectively. The temperature of the FID detector was set at 300 °C.

Data processing was performed using MSD ChemStation (revision F.01.00.1903) in combination with Automatic Mass Spectral Deconvolution and Identification System (AMDIS) (revision 2.70) and the National Institute of Standards and Technology (NIST) MS Search (version 2.0g) software (Agilent Technologies, Santa Clara, CA, USA). The identification of compounds was based on the comparison of the Electron Ionization (EI) mass spectra of the volatiles with data from the NIST11 and a mass spectral database of 567 pesticides and endocrine disrupters generated under Retention Time Locking conditions (RTLPEST 3 mass spectra libraries) with a probability of more than 50%. The percentage composition of particular component in the irradiated liquid material was determined on the basis of an area percent report (uncalibrated calculation procedure) generated by Agilent ChemStation software.

## 3. Results and Discussion

### 3.1. Synthesis of Homopolymeric Poly(N-Isopropylmethacrylamide) Hydrogel

The thermally-initiated free radical polymerization method to synthesize a series of poly(*N*-isopropylmethacrylamide) homopolymer hydrogels with 1.5, 2.0, 2.5, and 3.0 mol% of EGDM cross-linkers was applied. The electrostatic attraction between nucleophilic nitrogen atoms and the hydrogen atoms, as well as with the carbonyl group from *N*iPMAm and EGDM, can lead to the formation of intramolecular and intermolecular hydrogen bonds. The p(*N*iPMAm) homopolymer conformation depends on intramolecular interactions (hydrogen bonds, dipole–dipole interactions, and hydrophobic and electrostatic interactions). The chemical reactions of the cross-linked p(*N*iPMAm) homopolymer network with the indicated potential intramolecular hydrogen bonds formation is presented in Figure 1.

### 3.2. Structural Characterization of Synthesized Poly(N-isopropylmethacrylamide) Hydrogels

#### 3.2.1. FTIR Spectroscopy Analysis

Figure 2a shows the FTIR spectrum of the *N*-isopropylmethacrylamide monomer. In the wavenumber range of above 3000 cm^−1^, there was a characteristic high intensity absorption band with a maximum at 3292 cm^−1^ corresponding to N–H stretching vibrations, ν(N–H), that indicated the existence of a secondary amino group in the monomer structure. Next to this, an absorption band occurred at 3059 cm^−1^, which resulted from the asymmetric vibrations of the vinyl group, ν_as_( =C–H). The asymmetric and symmetric stretching vibrations of the C–H bond from methyl groups in the FTIR spectrum of *N*iPMAm gave bands with maxima at 2970 and 2876 cm^−1^, respectively. The medium intensity absorption band with a maximum at 2929 cm^−1^ was attributed to the asymmetric stretching vibrations of the C–H bond, ν_as_(C–H), from the isopropyl group of *N*iPMAm. The result of the vibrations of the bonds in the amide group of *N*iPMAm molecules was an amide band of high intensity with a maximum at 1651 cm^−1^ and an amide band II at 1538 cm^−1^. The amide band I was the result of C=O stretching vibrations, ν(C=O), while the amide band II was the result of the coupling N–H bending vibrations in the plane, δ(N–H) and the stretching vibrations of the C–N bond, ν(C–N). The existence of a double C=C bond in the *N*iPMAm structure was indicated by a medium intensity absorption band with a maximum at 1608 cm^−1^ that resulted from the stretching vibrations of the C=C bond, ν(C=C). The absorption band with a maximum at 1333 cm^−1^ also confirmed the presence of the amide structure, which is defined as the amide band III and which consequently led to the coupling of C–N stretching vibrations with N–H bending vibrations. A medium intensity absorption band at 1363 cm^−1^ corresponded to the bending vibrations in the plane of the C–H bond, δ(C–H), from isopropyl group in the *N*iPMAm structure. Bending vibrations in the plane of a vinyl group, δ( =C–H) in the FTIR spectrum of *N*iPMAm produced the band at 1465 cm^−1^, while out of plane bending, γ(=C–H), whose wavenumber and position depend on the substitution of the double bond, produced bands at 940 and 885 cm^−1^. In the wavenumbers range of 1000–650 cm^−1^, bands of C–H bending vibrations out of the γ(C–H) plane of the isopropyl group occurred [44]. Additionally, in this wavenumber range, the bending vibrations out of the γ(N–H) plane from the N–H group occurred, and in the FTIR spectrum of *N*iPMAm, the bands with maxima at 833 and 658 cm^−1^ originated from these vibrations.

The FTIR spectrum of EGDM presented in Figure 2b shows bands with the maxima at 2894 cm^−1^ originating from ν_s_(CH_3_), at 2960 cm^−1^ from ν_as_(CH_3_), at 2930 cm^−1^ from ν_as_(CH_2_), and at 3106 cm^−1^ originating from the vinyl group ν_as_(=CH). In this spectrum, there were also bands characteristic for ester and vinyl functional groups. The characteristic absorption band with the maximum at 1725 cm^−1^ in the FTIR spectrum of EGDM was assigned to C=O stretching vibrations of ν(C=O), which were conjugated with C=C bonds. The stretching vibrations of the dC–O bond produce a band with a absorption maximum at 1154 cm^−1^. The absorption band with the maximum at 1636 cm^−1^ originated from the absorbance of the C=C bond [44].

In the FTIR spectra of Figure 2c, a series of synthesized poly(*N*-isopropylmethacrylamide) homopolymer hydrogels with 1.5, 2.0, 2.5, and 3.0 mol% of EGDM are presented. The absence of some characteristic absorption bands, presented in the FTIR spectra of *N*iPMAm and EGDM molecules, indicates the formation of new structures. In FTIR spectra of p(*N*iPMAm), there were no bands from the double C=C bond and from the vinyl group, which indicates that the bonding of the *N*iPMAm monomers and cross-linking with the EGDM was made by the breaking of the C=C bonds. A wide band in the wave range of 3100–3700 cm^−1^ with the maximum at 3460 cm^−1^ originated from the stretching vibrations of the N–H groups, ν(N–H), in the *N*iPMAm monomer, indicating the formation of intramolecular hydrogen bonds between the hydrogel chains via an -NH group as the proton donor. The amide band II appeared at 1523 cm^−1^ with the maximum shifted by 15 units towards lower wavenumbers in relation to the same band position 5 in the FTIR spectra of *N*iPMAm. In the spectrum of p(*N*iPMAm), there was a low intensity band from the stretching vibrations of the C=O bond, ν(C=O), with a maximum at 1716 cm^−1^, which was shifted to higher wavenumbers by 9 units in relation to the position of the same band in FTIR spectrum of EGDM. A strong intensity amide band I appeared at 1648 cm^−1^, and it was shifted by 16 units to lower wavenumbers in relation to the position of the same band in the FTIR spectra of *N*iPMAm. Intramolecular hydrogen bonds could be formed via the C=O group from *N*iPMAm and EGDM and N–H group from *N*iPMAm. The shifting of these maxima towards lower wavenumbers indicated that the N–H and C=O groups of *N*iPMAm and EGDM participated in the hydrogen bond formation. The band from stretching vibrations of the CO group, ν_s_(C–O), with a maximum at 1157 cm^−1^, appeared in the FTIR spectrum of p(*N*iPMAm), and it was shifted by 4 units to higher wavenumbers in relation to the position in the EGDM. The discrete differences in the intensity and width of the absorption bands in the spectra of the synthesized homopolymers were due to the cross-linking reaction and the formation of hydrogels, as well as to the different mole fraction of the EGDM cross-linker. The results of the FTIR analysis are in agreement with the literature data [3,21,44,45].

#### 3.2.2. Scanning Electron Microscopy Analysis

To detect the morphologies of the p(*N*iPMAm) hydrogel sample with 3.0 mol% of EGDM, swollen in the equilibrium state, the micrograms were obtained by SEM (Figure 3). The pore size of the synthesized homopolymer p(*N*iPMAm) in the swollen state was up to 150 µm. Based on average pore size, the synthesized p(*N*iPMAm) can be classified as a macroporous hydrogel [2,28]. The three-dimensional structure of p(*N*iPMAm) hydrogels looked like a pretty settled, semi-uniform cross-linked network. This structural organization of p(*N*iPMAm) hydrogels provided a lot of free space within the cross-linked polymer network (shown in micro scale in Figure 3) between the nano-sized (or even smaller) polymer chains in the swollen state. For this reason, they can be applied for fluid sorption inside of the network or as carriers for many active substances.

### 3.3. Residual Reactant Analysis

The high pressure liquid chromatography method was used to determine the residual amount of the unreacted *N*iPMAm monomer in the synthesized p(*N*iPMAm) samples. Under the selected chromatographic conditions, a detection wavelength of 210 nm and a retention time (R_t_) of 2.506 min corresponding to the *N*iPMAm monomer were used. The HPLC chromatogram and UV spectrum of the *N*iPMAm standard solution, c = 0.257 mg/cm^3^, are shown in Figure 4. Calibration curve of p(*N*iPMAm) (Appendix A) and the areas maxima from the HPLC chromatograms (Appendix A) are shown in the Appendix A.

The results of the calculated amount of residual *N*iPMAm during the polymerization reaction, in relation to the total mass of homopolymer p(*N*iPMAm) xerogels and in percent in relation to the initial amount in the reaction mixture, are presented in Table 1 with peak area values.

The residual amount of the *N*iPMAm monomer in the p(*N*iPMAm) sample with 1.5 mol% of EGDM (59.023 mg/g or 6.039%) had a significantly higher value than other samples, and it was correlated with their gelatinous, paste-like appearance. The obtained values of the residual *N*iPMAm monomer amounts for the other p(*N*iPMAm) samples were in range of 22.616–35.774 mg/g, calculated per mass of p(*N*iPMAm) xerogels, or 2.367%–3.688% compared to the initial amount present in the reaction mixture. According to the standard (ISO 1567:1999), the maximum allowed amount of residual methyl methacrylate for hot polymerized acrylate was 2.2%, and for the cold polymerized, it was 4.5%. The acceptable limits for the amount of residual methyl methacrylate in dental materials should be in the range of 1%–3% [46]. These residual amounts of the *N*iPMAm monomer were within acceptable limits for the p(*N*iPMAm) samples with 2.0, 2.5, and 3.0 mol% of EGDM, because monomers are toxic in much higher amounts [43,44].

The analyzed methanol extract samples after synthesis did not show any residual amount of the cross-linker EGDM, which indicates that the total EGDM amount reacted during the p(*N*iPMAm) polymerization process. For this reason, the specific analysis was excluded from this representation. As a result of the obtained analysis, it was clear that the complete conversion of the monomer into the polymer was not achieved in the polymerization process, and there were some amounts of unreacted monomer in the polymerized p(*N*iPMAm) hydrogels. The presence of the residual *N*iPMAm monomer in the hydrogel samples was influenced by the physicochemical properties of the synthesized p(*N*iPMAm) samples, especially the sample with 1.5 mol% of EGDM in relation to the others in the synthesized series. A larger quantity of residual monomer indicated the weak bonding of the monomers into long chains, leading to the formation of a weak, gelatinous, paste-like structure that was different from the other samples that had a more solid, cross-linked structure.

### 3.4. Swelling Study

The behavior of the poly(*N*-isopropylmethacrylamide) xerogels that were swollen in distilled water was monitored at 20 °C and in the fluids with pH values 2.0 and 7.4. The swelling ratio, α, was calculated according to Equation (1), and the equilibrium swelling ratio, α_e_, was calculated according to Equation (2). The mass of the xerogel samples (m_0_) was measured until equilibrium (m_e_) was reached. As expected according to the previous analysis, the p(*N*iPMAm) sample with 1.5 mol% of EGDM decomposed during the swelling process in the tested fluids, so the results were dismissed from the analysis.

#### 3.4.1. Equilibrium Hydrogel Swelling at 20 °C and pH = 2.0

Changes in the swelling ratio, α, were calculated as the ratio of the absorbed mass of water and the mass of xerogels for the samples of the synthesized p(*N*iPMAm) hydrogels with 2.0, 2.5, and 3.0 mol% of EGDM at 20 °C in an acidic solution with pH = 2.0 depending on the swelling time and the EGDM; these results are shown in Figure 5a,b, respectively.

The p(*N*iPMAm) hydrogels achieved an accelerated swelling process during the first 190 min, reaching equilibrium after 360 min (Figure 5a). The p(*N*iPMAm) hydrogel sample with 2.0 mol% of the EGDM cross-linker achieved the highest swelling ratio (α_e_ = 21.77). The hydrogel with 3.0 mol% EGDM showed the lowest value of the equilibrium swelling ratio (α_e_ = 12.64). The influence of the EGDM cross-linker content on the swelling capacity of the p(*N*iPMAm) hydrogels (Figure 5b) showed a decrease in the swelling with the increase of the amount of cross-linker EGDM. The swelling curves of the p(*N*iPMAm) showed that the amount of cross-linker had an effect on the hydrogel swelling ratio. Because of the p(*N*iPMAm) hydrogel network with a higher cross-linker content, the polymer chains were more fixed and had less ability to absorb fluid. When there was a smaller amount of cross-linker, the length of the polymer chains between the two cross-linkage points was larger, and the network could expand more and achieve highest swelling ratio.

#### 3.4.2. Equilibrium Hydrogel Swelling at 20 °C and pH = 7.4

The dependence of the swelling capacity changed for the p(*N*iPMAm) hydrogels depending on the time and on the EGDM cross-linker contents at 20 °C in the fluid with a pH value of 7.4 are shown in Figure 6a,b, respectively.

All samples of the p(*N*iPMAm) hydrogels with varied cross-linker contents (2.0, 2.5, and 3.0 mol% of EGDM) in fluid with pH = 7.4 at 20 °C intensively swelled during the first 190 min, reaching a swelling capacity close to equilibrium state (Figure 6a). The p(*N*iPMAm) hydrogel sample with 2 mol% of EGDM achieved the highest equilibrium swelling ratio (α_e_ = 18.92). The hydrogel sample with 3 mol% EGDM swelled the least (α_e_ = 10.82). It could be seen that the amount of absorbed fluid depended on the cross-linker amount in the synthesized p(*N*iPMAm) hydrogels in a similar way as previous analysis in fluid with pH = 2.0. Hydrogel samples with fewer cross-linkers reached a higher swelling ratio value (Figure 6b) in a weakly alkaline medium (pH = 7.4). Swelling capacity increased with the decreasing of cross-linker content. The cross-linking degree of a polymer can be regulated applying a specified amount of cross-linker in the polymerization mixture. Based on free volume theory, the penetrant self-diffusivity can be easily predicted [47]. The total free volume, according to Vrentas-Duda, can be divided into interstitial free volume (which is not implicated in facilitating transport through the liquid) and the hole-free volume. It is presumed that the hole free volume dictate molecular transport and can be predicted according to the type of absorbed fluid.

An analysis of the obtained results showed that hydrogels exhibited a slightly higher swelling ratio in acidic fluid in relation to the weakly alkaline fluid (from α_e_ = 21.77 at pH = 2.0 to α_e_ = 18.92 at pH = 7.4 for the sample with 2 mol% of EGDM). The weak pH responsiveness in the surrounding fluid of p(*N*iPMAm) homopolymers was visible in real systems during the swelling study, as opposed to the expected behavior, because they are not classified as pH-sensitive. It is known that pH-sensitive cationic hydrogels swell more intensively in acidic fluids, indicating that protonated nitrogen from the amide group in the side chains of p(*N*iPMAm) hydrogels might be responsible for the weak pH sensitivity. The amide groups could be as hydrogen bond donors and acceptors, but because of the conjugation with the C=O group, the nitrogen atom could only be as a weak π-acceptor of the hydrogen bond. These weak interactions were characterized by a larger equilibrium distance between donor and acceptor compared to strong hydrogen bonds. As a result, the secondary amino group in the amide bond was difficult to protonate due to the presence of the adjacent carbonyl group. However, in the presence of fluids with different pH values, they became primarily ionized and could provide positive charges for ionic interactions with negatively charged functional groups at small distances. The functional groups responsible for the intermolecular associations in amides were the amino and carbonyl linkages, forming hydrogen bonding of the type N–H⋅⋅⋅⋅O=C (as is indicated in Figure 1) [48]. In the presence of acidic fluid (pH = 2.0), the secondary amino groups of *N*iPMAm could exist in the protonated state (NH^2+^), which could then lead to the electrostatic repulsion of polymer chains and a slightly higher swelling capacity compared to swelling in the fluid with a higher pH value (pH = 7.4). Synthesized p(*N*iPMAm) hydrogels exhibited weak pH sensitivity, i.e., they were not significantly sensitive to the pH value changes, similar to that described in the literature [44]. The protonation of the carboxamide nitrogen atom is an essential part of in vivo and in vitro processes (cis–trans isomerization, amide hydrolysis, etc.) [49]. This phenomenon has been well studied in geometrically strongly distorted amides, although there are little data concerning the protonation of undistorted amides. The formation of a hydrogen bond with weakly distorted amide nitrogen can, in general, be regarded as the incipient state of a proton transfer process. Several studies supporting unusual N protonation and amide functionality have been reported [50]. Dybal and co-workers calculated model structures and illustrated two essential hydrogen bonding interactions in polymer solutions: hydrogen bonds between close amide groups and hydrogen bonds between amide groups and hydrating water molecules. These calculations indicated that hydrogen bonds between N–H groups and water molecules are stronger than bonds between two amide groups as the result of the lengths of the hydrogen bond C–O⋅⋅⋅H–N [21].

#### 3.4.3. Swelling Kinetic Analysis

In order to study the nature of the diffusion process for the fluid within p(*N*iPMAm) hydrogels with different cross-linker contents, Fick’s Equation (4) was applied to fit and analyze the experimental data [44]:(4)$$F=\frac{M_{t}}{M_{e}}={\mathrm{kt}}^{n}$$
where *F,* i.e., *M_t_/M*_e_, is the fractional sorption; *M_t_* is the mass of the absorbed fluid at the time *t*; *M*_e_ is the mass of the absorbed fluid at the equilibrium state; *k* is the incorporated constant characteristic for the specific polymer network; and *n* is the diffusion exponent. The exponents *n* and *k* are values determined from the slope and intercept of the plots of ln *Mt*/*Me* versus ln*t* for p(*N*iPMAm) hydrogels. If *n* is less than 0.5, the swelling process is controlled by the Fickian diffusion mechanism. If *n* is between 0.5 and 1.0, the diffusion and polymer chain relaxation control the swelling process and indicates an anomalous diffusion mechanism known as non-Fickian diffusion. The values of *n* greater than 1.0 are described as type III (Case III) or Super Case II. The calculated values of diffusion exponent, *n* and constant characteristic, *k*, specific to test the swelling behavior of the p(*N*iPMAm) hydrogels at 20 °C and pH 2. 0 and 7.4, are shown in Table 2.

The calculated values of diffusion kinetic exponent *n* were in range of 0.077–0.302 at 20 °C for both fluids, with pH = 2.0 and pH = 7.4 and a dependence of *n* versus EGDM cross-linker content, according to p(*N*iPMAm) hydrogel swelling, is presented in Figure 7. These results proved that the swelling transport mechanism was the Fickian type diffusion, and the swelling process was controlled by diffusion.

The most often used method for determining the fluid diffusion coefficient *D* into hydrogel assumes thin plate geometry, only considering the initial phase of swelling (60% swelling) during which the thickness of the sample basically remains constant [44]. For the Fickian diffusion, the constant *k* is then related to the diffusion coefficient and the diffusion coefficient *D*, as calculated from Equation (5):(5)$$\frac{M_{t}}{M_{e}}=4{\left(\frac{Dt}{\pi l^{2}}\right)}^{1/2}$$
where *l* is the thickness of the sample in xerogel state. By applying a logarithm to Equation (5), a straight line dependence between ln (*M*t/*M*e) and ln *t* can be obtained. The calculated values of the diffusion coefficient, *D*, for the p(*N*iPMAm) hydrogels at 20 °C and pH = 2.0 and pH = 7.4 are included in Table 2. The presented data show the diffusion coefficient of the p(*N*iPMAm) hydrogels in the strongly acidic fluid (in range of 7.423–8.083⋅10^−4^ cm^2^/min) and show an increase in values for all samples of hydrogels, i.e., a faster diffusion compared to the diffusion process in a weakly alkaline solution (in the range of 3.403–5.115⋅10^−4^ cm^2^/min).

#### 3.4.4. Thermosensitivity Testing

The thermosensitivity testing of the synthesized poly(*N*-isopropylmethacrylamide) hydrogels was monitored throughout the change in the swelling ratio during the temperature increasing from 20 to 65 °C in fluids with pH = 2.0 and pH = 7.4, and these results are shown in Figure 8a,b, respectively.

The highest values of the equilibrium swelling ratio were reached by the p(*N*iPMAm) hydrogels at a lower temperature (20 °C) (in fluid with pH = 2, α_e_ = 21.14, and in fluid with pH = 7.4, α_e_ = 18.21). An intense decline in the swelling ratio occurred when the temperature rose to 40 °C, and at about 45 °C, the dependence started asymptotically approaching a constant lowest value (in fluid with pH = 2, α = 1.29, and in fluid with pH = 7.4, α = 1.18). The most intense phase transition was observed in the 35–40 °C range (Figure 8). It was also observed that the swelling ratio steadily declined with the increase in the cross-linking degree, as expected. The higher content of the cross-linker in the hydrogel sample increased the cross-linkage density of the network, reducing the mobility of polymer chains and the elasticity of the network. The results indicated that the synthesized hydrogels were negatively thermosensitive and had a LCST. At lower temperatures, hydrogen bonds between the water molecules and hydrophilic parts of the polymer chain were dominant. When the temperature increased, the average kinetic energy of the molecule also increased and the hydrogen bond weakened. The influence of the hydrophobic part in the polymer molecule prevailed, and, because of that, the molecule contracted.

### 3.5. Gel Permeation Chromatography Analysis of Hydrogel after Radiolysis

The GPC method allowed for the determination of the molar mass distribution of the liquid sample of the synthesized p(*N*iPMAm) hydrogel with 1.5 mol% EGDM after gamma-ray irradiation. The change of signal from the RID detector for the irradiated liquid material, depending on the eluted volume, is shown in Appendix A. The differential curve of the molar masses distribution is shown in Appendix A. Values of number average molar mass (Mn), mass average molar mass (Mw), Z average molar mass (Mz), and polydispersity index for irradiated liquid material from elugrams for fractions in the range of 5.30–7.60 min are presented in Table 3, and the accompanying calibration curve, elugrams, and differential curves of the molar masses distribution are available in the Appendix A (Appendix A).

The results of the GPC analysis indicated that there were no molecules of higher average molar masses in the liquid sample of hydrogels after irradiation with gamma rays; additionally, after dissolution and filtration, only monomers and oligomers were present. These results indicated the decomposition process after the gamma irradiation of the synthesized p(*N*iPMAm) hydrogel with 1.5 mol% EGDM and the scission of some of the cross-linked structures.

### 3.6. HSS-GC/MS and HSS-GC/FID Analysis of Hydrogel Sample after Radiolysis

The qualitative analyses of the isolated molecules were carried out using gas chromatography–mass spectrometry (GC–MS), while the quantitative analysis was achieved with gas chromatography combined with flame detection (GC–FID). The results obtained by this analysis showed the presence and content of 10 different chemical compounds identified in the liquid material after the irradiation of the p(*N*iPMAm) hydrogel (Figure 9 and Table 4). After the deconvolution of the co-eluting peaks under broad signals (peaks 1 and 2) in AMDIS, the created pure spectra for each component are shown in the Appendix A.

Corresponding to the products obtained during thermal decomposition, the most dominant compounds present at different percentage on the GC–MS chromatogram (Table 4) were: heptane (1.9%–77.4%), ethylisobutyrate (13.9%–42.4%), *N*-isopropylcyclopropanecarboxamide (0.2%–29.9%), and 2,2’-azobis(2-methylpropionitrile) (2.5%–18.3%). The most dominant volatile compound at 60 °C was heptane (76.3%), together with ethylisobutyrate (13.9%) and 2,2’-azobis(2-methylpropionitrile) (8.3%), as well as negligible amounts of *N-(n*-propyl)acetamide (0.7%), cyanuric acid (0.4), and 1-methyl-1,2,4-triazole (0.2%). When temperature increased to 90 °C, the composition of the most represented compounds was similar: heptane (77.4%), ethylisobutyrate (19.2%), 2,2’-azobis(2-methylpropionitrile) (2.5%), and 1-methyl-1,2,4-triazole (0.7%), with the appearance of a new molecule of 2-hydroxypropanenitrile (0.2%). The volatile component content changed remarkably when the temperature rose to 120 °C: heptane (49.3.4%), ethylisobutyrate (42.4%), 2,2’-azobis(2-methylpropionitrile) (5.2%), *N*-isopropylcyclopropanecarboxamide (0.7%), 2-hydroxypropanenitrile (0.2%), and 1-methyl-1,2,4-triazole (0.2%). At the significant thermal change at 150 °C, the volatile compound composition was transformed with the appearance of new molecules (nine identified in total, accounting for 97.5%). It is important to notice the rapid increase in content of *N*-isopropylcyclopropanecarboxamide (29.9%), 2,2’-azobis(2-methylpropionitrile) (18.3%), and ethylisobutyrate (38.9%), as well as the repeated appearance of *N-(n*-propyl)acetamide in a significantly higher amount (5.1%).

In the available literature, the authors of this manuscript did not find the results of p(*N*iPMAm) thermal degradation. The results of poly(*N*-isopropylacrylamide) pyrolysis (by Py-GC/MS analysis at 600 °C) included the following compounds identified in the program: 1-propenamine (8.44%), *N*-(1-methylethyldiene)-2-propanamine (7.40%), and 2-propanamine (1.36%) [51]. There were no identical compounds in the results obtained in the presented work.

The initiator 2,2’-azobis(2-methylpropionitrile) is thermally unstable, with an auto-ignition temperature of 64 °C and a melting process accompanied by simultaneous decomposition (mp 102–104 °C (dec.). It is interesting to note that after gamma-ray irradiation, 2,2’-azobis(2-methylpropionitrile) existed at a raised temperature (from 8.3% at 60 °C to 18.3% at 150 °C) (Table 4). A recrystallization process with methanol has been described in the literature [52]. It is possible that a recrystallization process in the presence of ethanol occurred. It is probably that the absorption of gamma-ray additionally stabilized this initiator. In this context, it will be useful to investigate the recyclability of obtained liquid material for new hydrogel synthesis, as well as their new properties. Based on the presented results of the HSS-GC/MS and HSS-GC/FID analysis of the liquid material after the irradiation of p(*N*iPMAm) hydrogels, its decomposition and the formation of new molecules apparently occurred. Given that the mechanism of degradation was not yet completely clear, an analysis of the possible mechanism of the formation of the four main compounds—*N*-isopropylcyclopropanecarboxamide, heptane, ethylisobutyrate, and *N-(n*-propyl)acetamide—was performed in this study. Potential pathways for γ-irradiation-induced decomposition of the p(NiPMAm) hydrogels with the chemical structures of the major volatiles (present in the headspace) are presented in Figure 10.

The formation of the major compounds could have occurred through a mechanism based on the radical degradation of the polymer. When the p(*N*iPMAm) hydrogel in the gelatinous form was irradiated by a γ-ray source, two reactions were likely to occur: the scission of the main chain and the decomposition of the side chains with gas evaporation.

The radiolysis of the ethanol, present as the solvent, created the initial primary free radicals, which then reacted with the polymer with the separation of hydrogen. The generation of free radicals in the radiolysis of ethanol can be represented by Reaction (6), followed rapidly by Reaction (7):

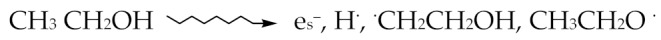
(6)
H**^⋅^**/CH_3_CH_2_O **^⋅^ +** CH_2_CH_2_OH → **^⋅^** CH_2_CH_2_OH + H_2_/CH_3_CH_2_OH(7)

This system generated the reducing radical CH_3_CH_2_O^⋅^ [35]. After the initial ionization, the released primary electrons moved through the polymer network and caused further secondary ionization and excitation. Other electrons, ions, excited parts of the molecules, and free radicals were formed, and these further reacted with each other and with the present structures, creating stable products. The additional hydrogen atoms came from generated products after the radiolysis of the ethanol, as presented in Reactions (6) and (7), and these then reacted with the polymer structures obtained after the scission of the main chain creating stable products. The degradation of the polymer network was likely to occur through the random scission of the C–C bonds in the base chain and the rearrangement of the excited polymer network. Firstly, secondary radicals were created through the scissoring of the C–C bonds. Due to the scission of the main chain, the part with seven C-atoms reacted with abstracted, mobile six H**^⋅^** species, thus resulting in the formation of a new structure identified by HSS-GC/MS and HSS-GC/FID analyses in the form of aliphatic hydrocarbons, such as heptane and related compounds. As a consequence of the breakup of the side chains, the residues of the monomer units and the residues of the EGDM cross-linker were separated. With the scission, the EGDM residue reacted with the mobile hydrogen atom leap (3H**^⋅^**), and a relatively stable new ethyl isobutyrate entity was formed. The subsequent rearrangement of the residues of the monomer unit of *N*-isopropylmethacrylamide, after the separation from the main chain, resulted in the cyclization of that part of the structure (Figure 10) and the formation of a new compound of *N*-isopropylcyclopropanecarboxamide, which was also identified by analysis. After the separation from the main chain and the rearrangement of the *N*-isopropylmethacrylamide residues and reaction with mobile three hydrogen atoms, the *N-(n*-propyl)acetamide was formed, though this occurred in a higher amount when the temperature rose to 150 °C. According to the literature data, the random main chain scission was accompanied by a change in the viscosity [53,54], whereby the molecular weight of the irradiated polymer was reciprocal to the radiation dose, and 61 eV was absorbed per the fractured main chain C—C bond. The radiation-induced degradation yield of polyisobutylene was studied at temperatures between 63 and 343 K. A decrease in the activation energy occurred at 83 K was assumed to accelerate monomolecular rupture processes involving the detachment of methyl groups [55].

## 4. Conclusions

The synthesis of p(*N*iPMAm) hydrogels was performed, and the structure was confirmed by the analysis of FTIR spectra. SEM micrographs indicated a semi-uniform cross-linked network, suggesting that p(*N*iPMAm) hydrogels can be classified as macroporous. The swelling behavior of this negatively thermosensitive hydrogel showed a slightly higher swelling ratio at 20 °C in fluid with pH = 2.0 (α = 21.14), in relation to the weakly alkaline conditions. The swelling transport mechanism was the Fickian type diffusion, and the swelling process was controlled by diffusion. The diffusion coefficient, D, of the p(*N*iPMAm) hydrogels in the strongly acidic fluid showed a faster diffusion compared to the diffusion process in a weakly alkaline solution. The residual amount of the *N*iPMAm monomer was within acceptable limits for p(*N*iPMAm) samples with 2.0–3.0 mol% of EGDM. The p(*N*iPMAm) hydrogel morphology provided a lot of free space within the cross-linked polymer network between the polymer chains in the swollen state, and, as such, they could be applied for fluid sorption or as carriers for many active substances.

The hydrogel samples in the gelatinous state were subjected to gamma irradiation in order to additionally initiate and form a solid three-dimensional structure. After an absorption irradiation dose of 31 kGy from the gamma source, the synthesized gelatinous samples unexpectedly transformed into the liquid phase. The p(*N*iPMAm) hydrogel with 1.5 mol% of EGDM after irradiation was not additionally cross-linked, but, contrary to expectations, its decomposition occurred, as confirmed by the results of GPC, HSS-GC/MS, and HSS-GC/FID. The four dominant compounds after irradiation: were found: *N*-isopropylcyclopropanecarboxamide, ethylisobutyrate, *N-(n*-propyl)acetamide and heptane, which were different from the reactants (monomer and cross-linker) and from the homopolymer. These findings indicate that gamma irradiation will be applicable for the hydrogel decomposition. However, the decomposition protocol using gamma irradiation should be further optimized. In future research, it will be useful to investigate the recyclability of the obtained liquid material for new hydrogel synthesis and their new properties.

## Figures and Tables

**Figure 1 polymers-12-01112-f001:**
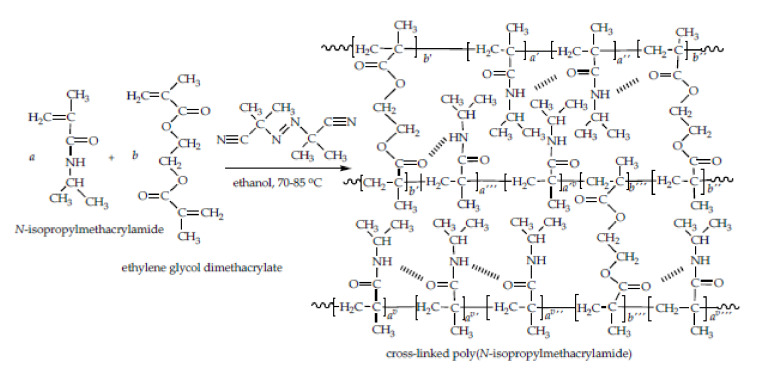
The chemical reaction of the cross-linked poly(*N*-isopropylmethacrylamide) (p(*N*iPMAm)) homopolymer network synthesis with an indication of the potential intramolecular hydrogen bonds formation.

**Figure 2 polymers-12-01112-f002:**
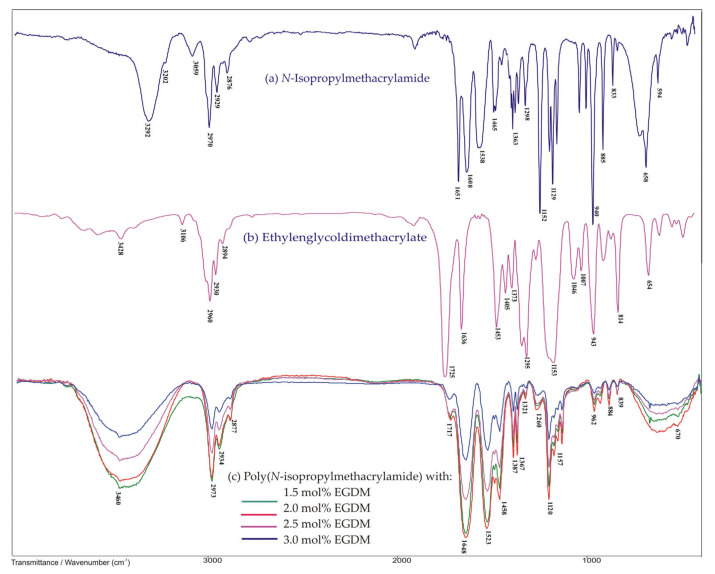
FTIR spectra: (**a**) *N*-isopropylmethacrylamide, (**b**) ethylene glycol dimethacrylate, and (**c**) synthesized poli(*N*-isopropylmethacrylamide) samples with 1.5, 2.0, 2.5, and 3.0 mol% of ethylene glycol dimethacrylate (EGDM).

**Figure 3 polymers-12-01112-f003:**
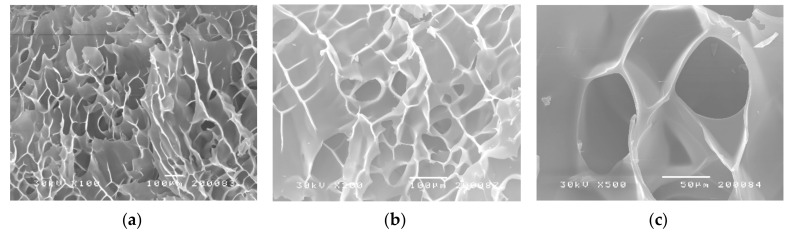
SEM micrographs of p(*N*iPMAm) hydrogel sample with 3.0 mol% of EGDM, swollen in the equilibrium state with magnification: (**a**) 100× (scale bar 100 µm), (**b**) 200× (scale bar 100 µm), and (**c**) 500× (scale bar 50 µm).

**Figure 4 polymers-12-01112-f004:**
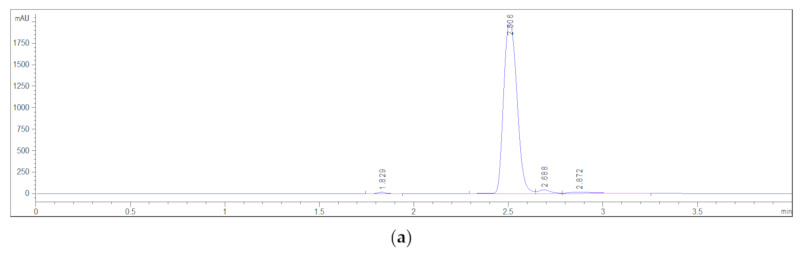
(**a**) HPLC chromatogram of standard solution *N*iPMAm, c = 0.257 mg/cm^3^; (**b**) UV spectrum of *N*iPMAm.

**Figure 5 polymers-12-01112-f005:**
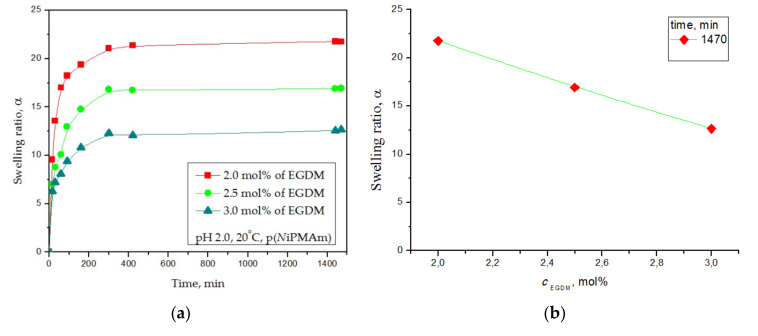
Swelling ratio, α, of the poly(*N*-isopropylmethacrylamide) hydrogels at 20 °C and pH = 2.0 depending on (**a**) time and (**b**) EGDM content.

**Figure 6 polymers-12-01112-f006:**
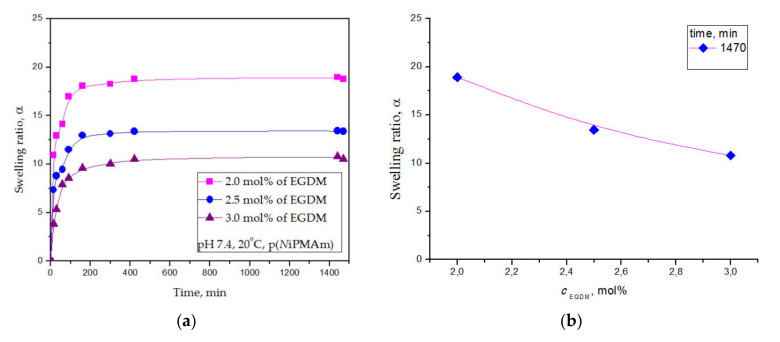
Swelling ratio, α, of the poly(*N*-isopropylmethacrylamide) hydrogels at 20 °C and pH = 7.4 depending on: (**a**) time and (**b**) EGDM content.

**Figure 7 polymers-12-01112-f007:**
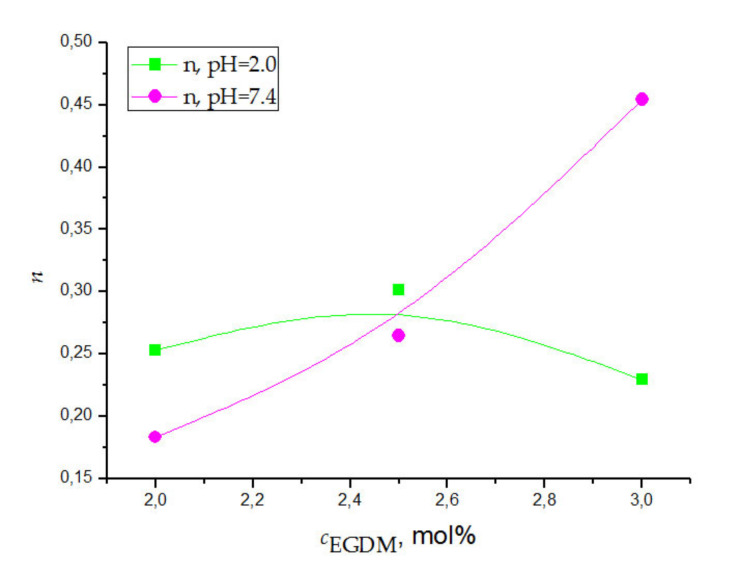
Dependence of *n* versus EGDM cross-linker content, according to p(*N*iPMAm) hydrogel swelling in fluid pH = 2.0 and pH = 7.4 at 20 °C.

**Figure 8 polymers-12-01112-f008:**
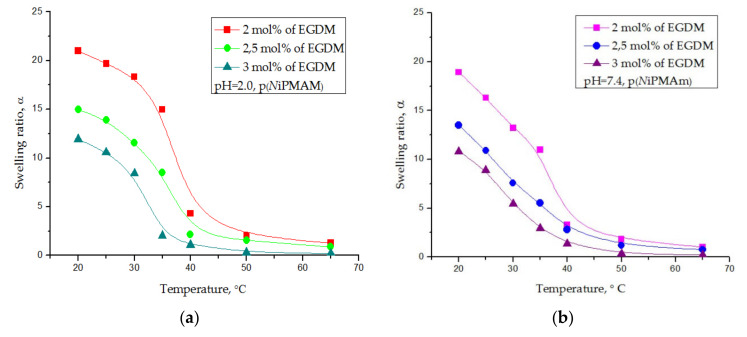
Swelling ratio, α, for poly(*N*-isopropylmethacrylamide) hydrogels depending on the temperature in fluid with pH-values of (**a**) 2.0 and (**b**) 7.4.

**Figure 9 polymers-12-01112-f009:**
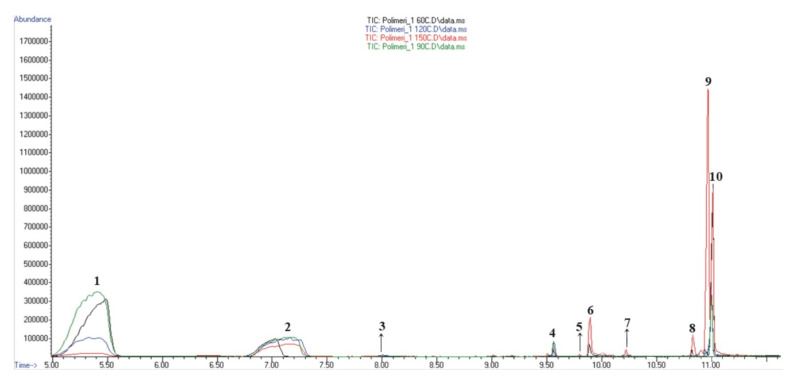
Comparative view of total ion chromatograms (TIC) of liquid material from irradiated p(*N*iPMAm) hydrogel volatiles obtained under different oven temperatures.

**Figure 10 polymers-12-01112-f010:**
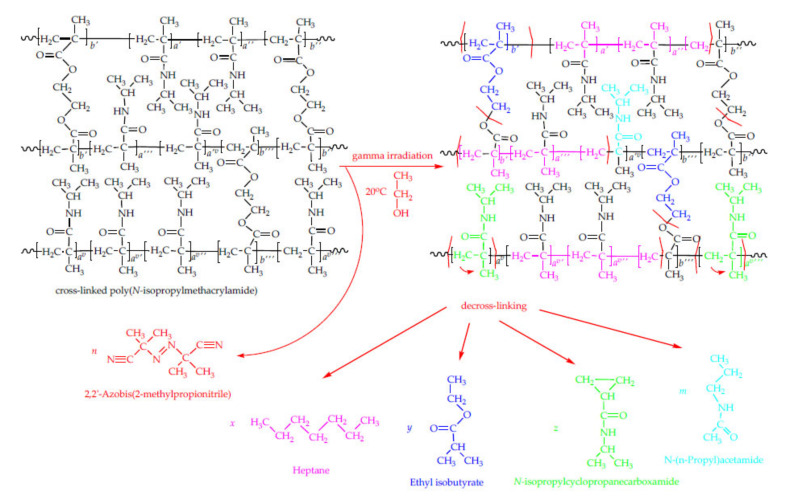
Potential pathways for the γ-irradiation-induced decomposition of the p(*N*iPMAm) hydrogels with the chemical structures of the major volatiles.

**Table 1 polymers-12-01112-t001:** The peak area values of the p(*N*iPMAm) xerogel samples masses and the residual amount of *N*iPMAm calculated in relation to the total mass of homopolymer p(*N*iPMAm) xerogels and in percent in relation to the initial amount in the reaction mixture.

p(*N*iPMAm), mol% of EGDM	Mass, g	*A*, mAU·s	*N*iPMAm, mg/g	*N*iPMAm, %
1.5	0.115	9166.1	59.023	6.039
2.0	0.207	9932.9	35.774	3.688
2.5	0.258	10138.7	29.455	3.059
3.0	0.357	10743.9	22.616	2.367

**Table 2 polymers-12-01112-t002:** The calculated values of kinetic parameters (*n*, *k*, and *D*) of fluid (pH 2.0 and 7.4) diffusion into p(*N*iPMAm) hydrogels at 20 °C.

p(NiPMAm), mol% of EGDM	pH 2.0				pH 7.4			
	*n*	*k*, min^−1/2^	*D*⋅10^4^, cm^2^/min	*R* ^2^	*n*	*k*, min^−1/2^	*D*⋅10^4^, cm^2^/min	*R* ^2^
2.0	0.253	0.250	7.423	0.950	0.183	0.249	5.115	0.966
2.5	0.302	0.184	7.591	0.991	0.265	0.218	3.716	0.996
3.0	0.229	0.262	8.083	0.991	0.454	0.077	3.403	0.972

**Table 3 polymers-12-01112-t003:** Values of average molar mass (*M*n), mass average molar mass (*M*w), *Z* average molar mass (*M*z), and polydispersity index (*D*) for irradiated liquid material from elugrams.

Elution Time, min	$\overset{-}{M}n,\;g/\mathrm{mol}$	$\overset{-}{M}w,\;g/\mathrm{mol}$	$\overset{-}{M}z,\;g/\mathrm{mol}$	D	Figure
5.30–7.60	1.148⋅102	2.040⋅102	3.180⋅103	1.777	S3
5.30–5.80	7.040⋅102	7.324⋅102	7.645⋅102	1.040	S4
5.80–6.36	3.049⋅102	3.217⋅102	3.376⋅102	1.055	S5
6.36–6.85	1.155⋅102	1.215⋅102	1.275⋅102	1.052	S6

**Table 4 polymers-12-01112-t004:** Chemical composition of the volatiles present in the headspace of liquid material from irradiated p(*N*iPMAm) hydrogels.

No.	*t*ret., min	Compound	Molecular Formula	Method of Identification	Prob. (%)	Area, %			
						60 °C	90 °C	120 °C	150 °C
1.	5.03–5.62	Heptane and related compounds		MS	76.6	76.3	77.4	49.3	1.9
2.	6.76–7.34	Ethyl isobutyrate and related compounds		MS	78.3	13.9	19.2	42.4	38.9
3.	8.01	2-hydroxypropanenitrile	C3H5NO	MS	95.8	-	0.2	0.6	-
4.	9.51	1-methyl-1.2.4-triazole	C3H5N3	MS	66.2	0.2	0.7	0.2	0.6
5.	9.80	4-ethoxy-2-pentanone	C7H14O2	MS	97.0	-	-	-	0.2
6.	9.89	N-(n-propyl)acetamide	C5H11NO	MS	80.9	0.7	-	-	5.1
7.	10.23	2-pentanone	C5H10O	MS	53.1	-	-	-	0.5
8.	10.82	Cyanuric acid	C3H3N3O3	MS	83.2	0.4	-	-	2.1
9.	10.95	N-Isopropylcyclopropanecarboxamide	C7H13NO	MS	81.0	0.2	-	0.7	29.9
10.	11.01	2.2’-azobis(2-methylpropionitrile)	C8H12N4	MS	64.7	8.3	2.5	5.2	18.3
				Total identified (%)		100.0	100.0	98.4	97.5

*t*_ret._: Retention time; MS: constituent identified by mass-spectra comparison.

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
