# Peer review of "Intelligent Poly(N-Isopropylmethacrylamide) Hydrogels: Synthesis, Structure Characterization, Stimuli-Responsive Swelling Properties, and Their Radiation Decomposition"

_polymers, 2020, doi:10.3390/polym12051112_

Round 1

Reviewer 1 Report

This manuscript has clarified the crosslinked P(NiPAMM) hydrogel. The pH and temperature responsiveness have been clearly performed. However, P(NiPAMM) homopolymer does not possess pH sensitivity. The pH responsiveness might result from the segments of P(NiPAMM) copolymers. In my opinion, the pH responsiveness of the hydrogel should be illustrated or discussed in this study.

Reviewer 2 Report

This manuscript is written poorly and its scientific significance is not offered clearly. The detailed comments are listed below and the reviewed manuscript with marking is also supplied. Thus, this manuscript is not good to be accepted.

Academic issues:

  1. This manuscript is composed of different parts and those of synthesis, characterization and stimuli-responsive swelling are correlated. The synthesis method was common radical polymerization, the characterization was regular one and the thermal response of PNiPMA has been studied widely. Thus, the significance of radiation decomposition in this manuscript and its correlation with other parts should be emphasized but were not interpreted in believable way. “The aim of this work” should not be confined to “the synthesis” (Line 18).
  2. The analysis of unreacted monomer in the obtained hydrogel should be clearly stated, probably concerning with the applications.
  3. “As a new-style polymer material” (Line 40): responsive polymers and hydrogels are not of new-style, referring to their development.
  4. Subsection 2.1: was the monomer purified before the synthesis? If so, offer the information.
  5. Line 202: this dryness method is not good for the maintenance of hydrogel morphology.
  6. Subsection 2.3.3: why was “moistened filter paper” used to wipe excess water?
  7. What is “hot cell” (Line 256)? Specify the statement of “using spent 192Ir and 75Se originated from gamma cameras used in radiography”.
  8. “for the activation of kinetic chains” (Line 260) is suggested to change into “for the activation of reactive species”.
  9. GPC calibration: two kinds of standard polymers (PNiPMAm and s-PSt) were used. Thus, both of them have three molecular weights, is it right? Thus, should be two calibration curves built up? What supplier offered those standard polymers? What sulfation degree of s-PSt?
  10. What is “the valence vibrations”? Please use the regular phrases, such stretching vibration and bending vibration.
  11. Line 352: “is” should be deleted. Since the amide group comes from the monomer, “the formation” should be changed into “the presence”.
  12. SEM analysis (Page 10): it should be noticed that the pore size is stated to be micro-scale but the space between polymer chains is nano-sized or even smaller. Thus, the related description in Line 408-409 should be corrected.
  13. Table 1 and the sentence in Line 442: the weight percent of residual monomer is several percent and was stated to be within “acceptable limits”. If the hydrogels are used in biomedical fields, the residual percent is too high. Please specify it.
  14. what does “noting detected” (Line 445) mean? How does “physicochemical properties” influence on the “the presence of residual monomer? (Line 450)
  15. Line 467 and 468 as well as Line 486 and 487: what is the difference between “the highest swelling capacity” and “equilibrium” swelling capacity in this manuscript? Please offer the definition of this manuscript.
  16. The statement in Line 494 and 495 is difficulty to follow. The phrase of “free volume” in polymer physics is specially defined (please refer to one textbook).
  17. The secondary amino group in amide bond is difficulty to protonate (Line 498) since the adjacent carbonyl group.
  18. “the density of the network” (Line 553) should replaced with “the crosslinkage density of the network”.
  19. What is “kinetic energy”? (Line 557)
  20. The titles of Subsection 3.5 and 3.6 is suggested to change to those exhibiting “radiation” or “radiolysis”.
  21. The data in Table 3 seem so strange. There are four eluent signals for one radiolysis sample (p(NiPMAm) hydrogel with 1.5 mol% EGDM). Why? The data of molecular weight are too exact, being above the accuracy of GPC measurement.
  22. The analysis of TIC results (Figure 9) is questionable. Many signals are too broad, indicating the mixture of different compounds for those broad signals.
  23. Please specify if “the authors” (Line 612” are those of this manuscript or those of the reference.
  24. Line 617 and 816: although melting point is one parameter concerning with thermal stability, the decomposition temperature or rate constant is the key. This sentence has grammatical error. The sentence in Line 619 and 624 is also difficulty to follow.
  25. Figure 10 and the related discussion: one key issues should be emphasized. What does the additional hydrogen atoms come, comparing with the corresponding structure in the backbone of polymer? The radiolysis mechanism and its description is open to discussion.

Writing issues

  1. The manuscript title: “intelligent hydrogels poly(N-Isopropylmethacrylamide)” is suggested to change into “intelligent poly(N-Isopropylmethacrylamide) hydrogels” and “swelling properties” into “stimuli-responsive swelling properties”.
  2. The previous aim of gamma-ray radiation might be the further crosslinkage of obtained hydrogels (to further initiate additional cross-linking, Line 24) but the gamma-ray radiation caused the decomposition. This kind of description should appear in Abstract as well Conclusion.
  3. Abstract: there are too many references and some of them are not closely correlated to the topic of manuscript. Abstract should be focused on the manuscript research and shortened.
  4. Check the correction in “p(N-n-propylacrylamide), p(NiPMAm)” (Line 76).
  5. Check and revise the sentence in Line 82-84. This sentence is difficulty to follow and has grammatical error.
  6. Improper citation: “which are known for their high cost and toxic nature [42]” (Line 135) is not consistent with “allows the fabrication of pure product non-contaminated with residuals of toxic initiators” (P479, ref 42). Please check other improper citations.
  7. The sentence in Line 143 and 144 was not clearly stated.
  8. Subsection 2.3.4: the statement of “being treated with of synthesed hydrogels during 36 h with stirring” may be wrongly stated.
  9. Line 319-326: it is not necessary to describe the polymerization mechanism as done in the sentences since it is common knowledge, even to one newcomer of polymer science.
  10. Line 376-383: those sentence is suggested to state concisely.
  11. The sentences in Line 420-423 as well as the description in Line 428-432 should be moved to Experimental section.
  12. The sentence in Line 613-615 is difficulty to follow.
  13. Suggested changes of writing: “transformed in” (Line 26) → “transformed into”; “gel permeable chromatography” (Line 26) → “gel permeation chromatography”; “Analysis the association and dissociation” (Line 63) → “Analysis of the association and dissociation”; “form larger aggregates” and “have” (Line 64) → “formws larger aggregates” and “had”; “reason of the p(NiPMAm)” (Line 92) → “reason that the p(NiPMAm)”; “hydrogel latexes prepared” (Line 115) → “hydrogel latexes were prepared”; “hydroxypropylmethacrylate” (Line 149) → “hydroxypropyl methacrylate”; “homopilymeric hydrogels” (Line 180) → “homo-polymeric hydrogels”; “Gamma source hydrogel irradiation” (Line 251) → “Gamma-ray irradiation of hydrogel”; “this complete transition” (Line 266) → “complete transition”; “for synthesis a series” (Line 317) → “to synthesize a series”; “originating from” (Line 362) → “originate from”; “absorption of the C=C bond” (Line 372) → “absorbance of the C=C bond”; “FTIR spectra a series of” (Line 373) and “was” (Line 374)→ “FTIR spectra a series of” and “are”; “formation of a of the homopolymer” (Line 474) → “”; “a snaller amount” (Line 476) → “a smaller amount”; “the two nodes” (Line 477) → “the two crosslinkage points”; “from” (Line 480 and 481) → “on”.
